# Design of a Tester for In Situ Simultaneous Measurement of the Wear of Two Different Film Materials

**Dongai Wang and Meihua Liu \***

Department of Mechanics, Tianjin University of Commerce, Tianjin 300134, China
* Correspondence: lmhua@tjcu.edu.cn

**Abstract:** The amount of wear is one of the most important indicators for assessing the wear resistance of materials, controlling product quality and studying the mechanisms of frictional wear of materials. Due to the limitations of the friction and wear test equipment currently in use for measuring material wear resistance, it is not possible to accurately compare the wear resistance of two different materials. To solve this problem, this paper proposes a new type of friction and wear tester. With the newly designed friction and wear tester, it is possible to perform friction tests and in situ measurements of wear on two different materials at the same time. This will significantly reduce the measurement errors of currently used friction and wear test equipment that requires adjustment in order to measure the amount of wear; it is particularly suitable for accurately comparing the wear resistance of two different materials. The newly designed friction and wear tester can be used extensively to test and analyse the wear resistance of solid materials such as metals, ceramic materials and engineering plastics.

**Keywords:** wear resistance; friction and abrasion tester; thin-film materials; in situ measurement; amount of wear and tear





## 1. Introduction

In mechanical equipment, friction exists between parts that have relative movement. Unusual wear and tear due to improper lubrication will lead to parts failing, thus affecting the normal operation of the equipment and also causing a huge waste of social wealth. It is therefore essential to study the wear resistance of materials [1,2]. The wear resistance of a material is measured by the amount of wear, which is an important indicator for assessing the wear resistance of materials, controlling product quality and studying the mechanism of friction and wear. This parameter of material wear can be obtained by measuring the change in length, weight or volume of the sample, referred to as length wear, weight wear and volume wear, respectively [3–6]. Length wear is determined by the amount of change in the normal dimension of the part's surface before and after the test, commonly measured by micrometres with an accuracy of 0.001 mm. This measurement method is often used when monitoring the wear status of parts in the actual working condition of machinery and equipment. Weight wear and volume wear are determined by the amount of change in the weight or volume, respectively, of the sample being measured before and after wear. One of the most common methods of measurement is the weighing method. This method of measurement usually uses a precision analytical balance with a measurement accuracy of 0.01 g to determine the amount of wear on a sample by weighing the change in mass of the part before and after wear. The method of determining the amount of wear by weighing is simple and easy to perform, which is why it is the most commonly used method. In order to minimise the effect of unclean surfaces on the weight of the sample being weighed, it is necessary to clean and dry the sample with an organic solvent, such as ethanol, before weighing it. However, materials with permeability such as epoxy resins are not easily cleaned because they tend to absorb oil during the friction test. This results in a large error in the determination of wear using the weighing method. Therefore, it is not advisable

to use the weighing method to determine the wear resistance of permeable materials. If the material under test has undergone a large plastic deformation of its friction surface during the friction process, there is a change in the size of the specimen, but the weight loss is not significant. It is also not possible to determine the amount of material wear by weighing in this case. The weighing method for determining wear is suitable for small test pieces and for materials that do not undergo large plastic deformations during the friction process. For materials with different densities, it is more reasonable to assess the degree of material wear in terms of volume wear rather than weight wear. Laboratory tests are often carried out by first measuring the weight wear of the specimen and then converting it to volumetric wear; the volume of wear can also be calculated by measuring the width of the abrasion marks, etc. In addition, there is also the use of electron microscopy and optical microscopy to observe changes in the metallographic microstructure of material surfaces before and after frictional wear experiments and to determine the degree of wear by analysing the pattern of change. This method is particularly suitable for the study of corrosion wear and fatigue wear.

Although the various methods of measuring material wear described above each have their own characteristics in terms of assessing the wear resistance of materials, friction tests on materials are often performed on different equipment to wear measurements. The error in measuring the amount of material wear in this way is often relatively large. The weighing method, in particular, is not suitable for accurate comparisons of the wear resistance of two different materials and is particularly unsuitable for research work on two different thin-film materials with small differences in wear resistance, because of the different measurement conditions [7–15]. As nanofriction continues to develop, the demands on tribological testing techniques are increasing. With this in mind, we designed a new friction and wear tester that can accurately measure the wear of materials in situ. This new friction and wear tester was designed to simultaneously perform friction tests on two different materials and then measure the length wear in situ. This ensures that the test conditions are the same, thus significantly reducing measurement errors due to movement. This friction and wear tester is particularly suitable for the accurate comparison of small differences in the abrasion resistance of two different film materials and can be widely used for testing the abrasion resistance of metal materials, ceramic materials, engineering plastics and other composite solid materials on their surfaces [16–23].

## 2. Presentation of the Problem

In order to investigate the effect of nanodiamond particles on the strengthening of nickel plating, a nickel-nanodiamond composite coating was prepared in the laboratory using a Watts plating solution. The sample specifications are shown in Figure 1 [24].

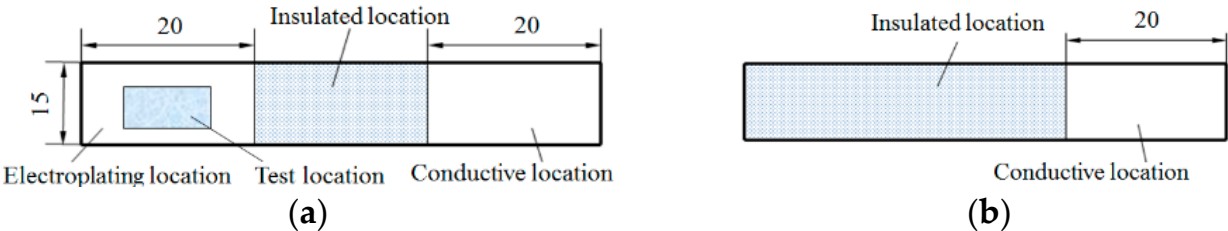

(a)　　　　　　　　　　　　　　　　　　　　　　　　　　(b)

**Figure 1.** Schematic diagram of the matrix size of the substrate and the sealing, plating and conductive locations: (**a**) front view and (**b**) back-face view.

The substrate material chosen was mild steel. In order to observe the internal organisation of the composite coating, a wire cutter was used to cut along the cross section of the sample. After rough and fine grinding of the cross section, the treated cross section was observed using a scanning electron microscope LEO1530-vp from Leo Electron Microscopy GmbH, Oberkochen, Germany. Figure 2 shows the cross-sectional profile of the nickel-nanodiamond composite coating obtained by electrodeposition. Figure 2a shows

the section after chemical corrosion and Figure 2b shows the section without corrosion. Figure 2c shows the elemental analysis of the section at different depths.

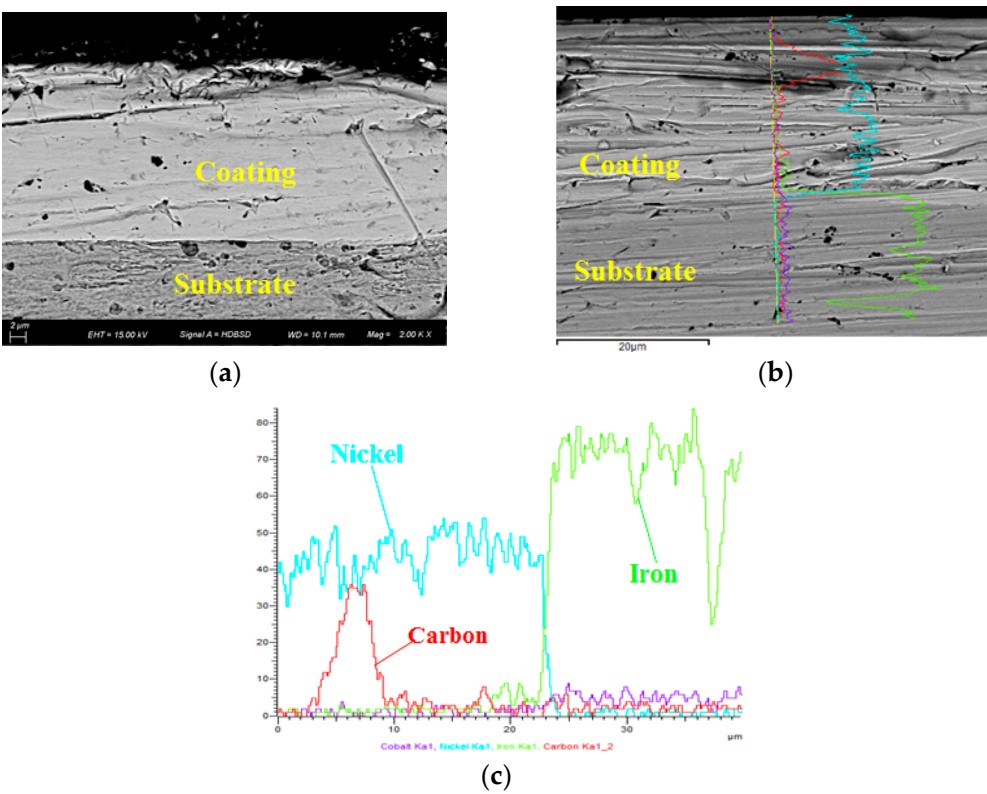

**Figure 2.** Cross-sectional image of the plating: (**a**) chemically etched cross section, (**b**) cross section before chemical etching and (**c**) analysis of cross-sectional elements at different depths.

In order to compare the wear resistance of the nickel-nanodiamond composite coatings prepared under different electrodeposition conditions, a microfriction tester, which is currently in common use on the market, was chosen for testing. The dyadic approach is a reciprocating movement of a ball and disc. The grinding tool is made of GCr15 bearing steel with a diameter of 8 mm. The experimental conditions were 5 N positive pressure, 120 r/min speed and 4 mm stroke. A scanning electron microscope LEO1530-vp from Leo Electron Microscopy GmbH, Germany was used to observe the abrasion pattern of the composite coating after the friction test. The results of the friction test are shown in Figure 3.

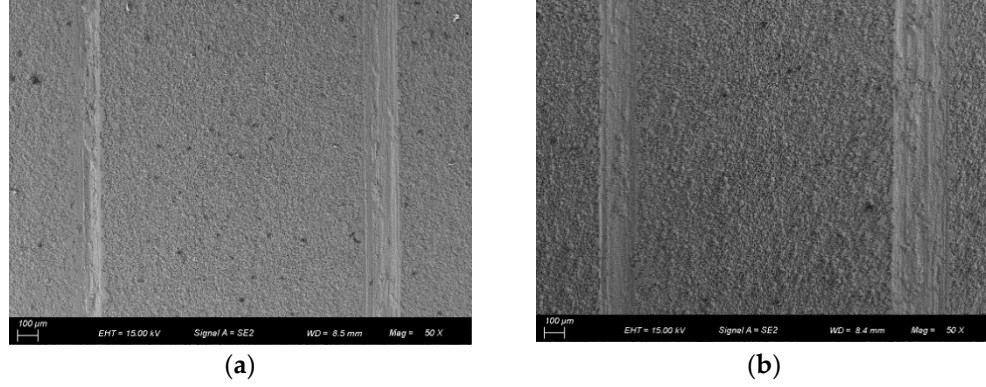

**Figure 3.** Frictional morphology of the composite coating prepared under two different electrodeposition conditions: (**a**) Sample 1 and (**b**) Sample 2.

It can be seen that when the same friction tester was used for the friction test, the widths of the two abrasion marks left on the surface of the same sample were clearly different. This is because existing friction testers can only be used for one friction test on the sample under test in one installation, i.e., only one abrasion mark can be obtained. In order to ensure that the experimental data were reasonable and that they reflected the performance of the sample under test, we carried out two abrasion friction tests on each sample under test. That is, after fixing the sample under test on the bench of the friction test machine, we started the friction test machine for testing. At the end of the experiment, the sample was released from the friction tester table. By visual inspection, the sample was repositioned on the table of the friction tester and fixed for the second friction test. Therefore, although the friction-testing machines currently in use play a positive role in the study of material friction properties, there are certain problems that need to be improved. The problem is particularly acute in the comparative study of two surface materials with small differences in friction properties [25].

## 3. Working Principle and Overall Solution Design

In order to solve the problems encountered in the above friction tests, we designed a new friction tester that can simultaneously perform friction tests on two test specimens. Figure 4 shows the appearance of the new friction and wear tester designed. In particular, Figure 4a shows an axonometric view of the appearance of the friction tester, and Figure 4b shows the front-view projection of the friction tester in the friction experimental condition. Before starting the frictional wear experiment, two different samples under test are mounted and fixed on the sample table (2), and the heights of each of the two samples are tested and recorded. After the height tests on the two different samples under test have been completed, the motor of the friction tester drives the grinding mechanism (1) to the friction test state, as shown in Figure 4b. At this point, the abrasive belt is driven by power to grind the sample. When the grinding is complete, the motor in the friction tester drives the grinding mechanism (1) away from the table (2) where the sample is mounted. At this point, the test system again tests and records the surface heights of the two samples being tested. The difference in height between each sample before and after the friction test is the amount of wear on the sample under test.

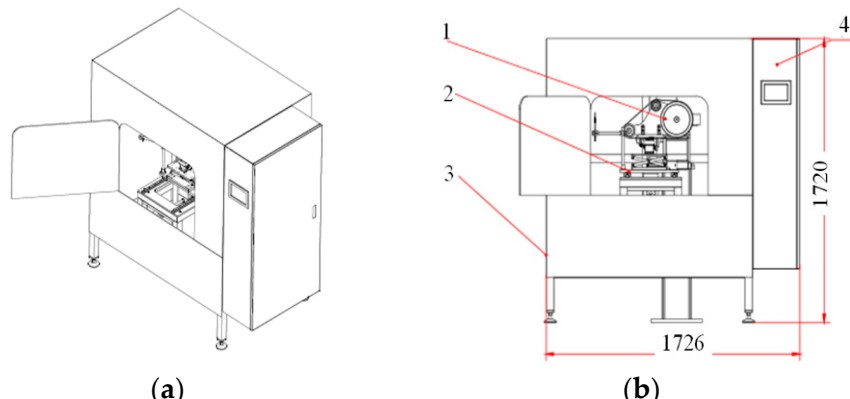

**(a)** **(b)**

**Figure 4.** Schematic diagram of the appearance of the frictional wear test equipment: (**a**) axonometric projection view of the appearance of the friction tester; (**b**) a front-view projection of the appearance of the friction tester. 1—grinding mechanism; 2—table for the sample to be tested; 3—outer cover of the tester; 4—control systems for equipment.

The mechanical part of the newly designed friction and wear tester consists of four main parts: the belt grinding mechanism (9), the table (4) on which the sample to be tested is mounted, the wear quantity testing mechanism and the base (1). Figure 5 shows a schematic diagram of the friction-testing machine performing grinding work. In particular, Figure 5a shows the complete working condition of the friction tester and Figure 5b shows

a partial enlarged schematic of the wear measurement process. The table on which the sample is mounted can be moved along guide rails (2) on the base, moving the table away from or closer to the grinding mechanism. At the same time, the table can be moved up and down by a lifting mechanism, by fast automatic or slow manual movement. The probe (7) for measuring wear can be rotated at an angle with the spindle (6) so that the probe (7) is in a position to measure the wear of the sample being measured or away from the sample being measured. To reduce the angular error caused by the rotation of the probe (7), the spindle (6) is supported by precision bearings in conjunction with the seat bore.

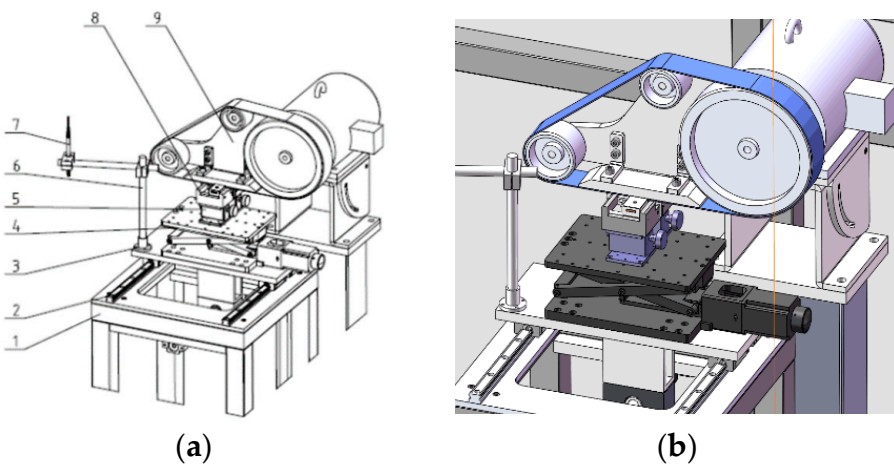

(**a**)  (**b**)

**Figure 5.** Schematic diagram of the friction test process of the test machine: (**a**) diagram of the complete friction-testing machine; (**b**) partial enlargement of the friction tester. 1—base; 2—guideway; 3—precision bearing; 4—table for the sample to be tested; 5—lifting mechanism; 6—spindle; 7—probe; 8—clamping mechanism; 9—belt grinding mechanism.

When the grinding mechanism has finished grinding, the motor drives the grinding mechanism to rotate 90° away from the surface of the sample to be measured. At this point, the probe (7) rotates at an angle with the spindle (6) and enters the surface of the sample to be measured, and the thickness of the sample is tested. This enables the in situ measurement of material wear. Figure 6 shows a diagram of the friction tester measuring wear in situ on the sample under test after the grinding work has been completed. In this case, Figure 6a shows the friction tester in a test condition for the amount of wear on the sample under test, and Figure 6b shows a partial enlarged schematic of the probe measuring the amount of wear on the sample under test.

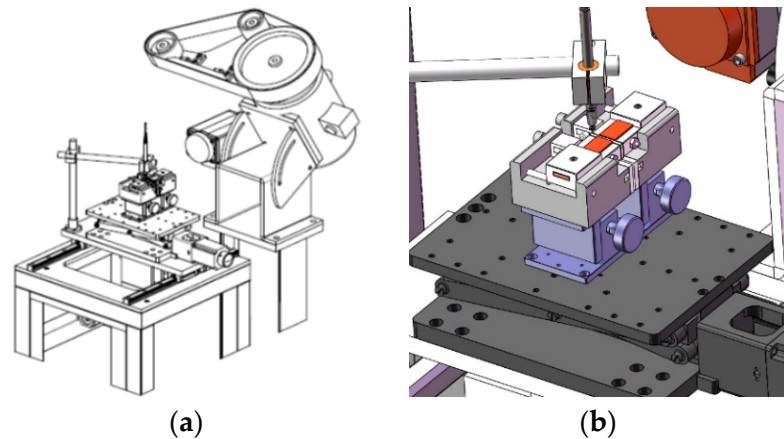

(**a**)  (**b**)

**Figure 6.** Schematic diagram of the friction tester measuring wear in situ: (**a**) schematic diagram of the complete appearance of the friction tester; (**b**) local enlargement of in situ measurement.

## 4. Design of the Main Components of the Test Machine

*4.1. Mounting Mechanism and Precision Adjustable Working Platform for the Sample to Be Tested*

The mounting mechanism and table designed for the sample to be tested are only suitable for mounting a specific size of coating. In practice, special fixtures can be designed to suit the specifications of the sample to be tested. The sample size used was 15~20 mm × 15 mm × 3 mm, and the surface film size was 10~15 mm × 15 mm × 20~30 μm (see Figure 7 for an illustration).

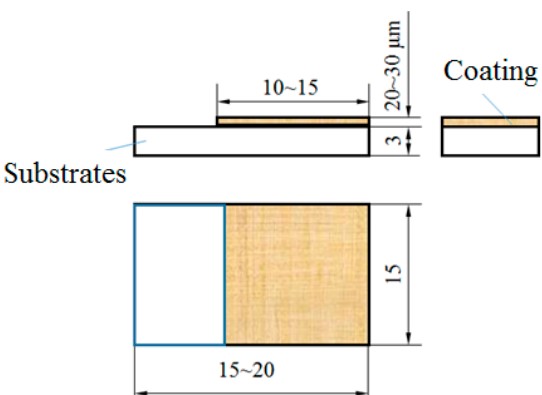

**Figure 7.** Specification of the sample under test and the dimensions of the surface film.

Figure 8a shows a schematic diagram of the mechanical mechanism, table and precision lifting mechanism for mounting the sample under test, and Figure 8b shows a partial enlargement of the mounting mechanism. As shown in Figure 8b, a temperature sensor and a force sensor are installed on the side and bottom of the sample under test. During the grinding operation of the friction tester, the sensors are able to capture the temperature change of the sample and the change in the amount of positive pressure and friction force applied in real time.

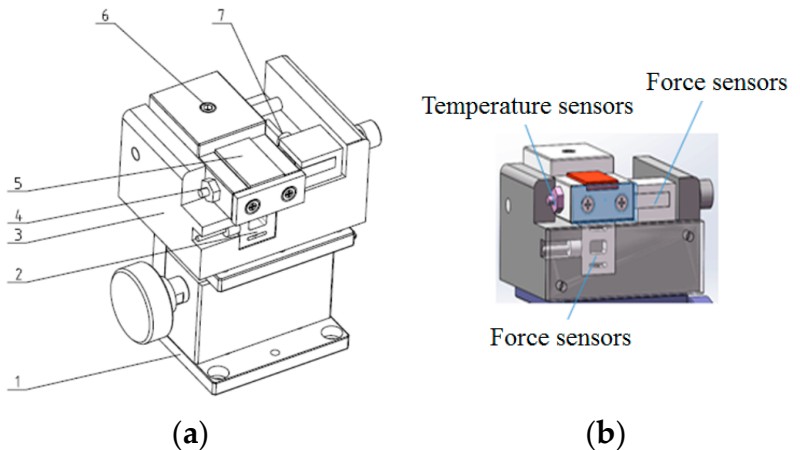

**(a)** **(b)**

**Figure 8.** Adjustable precision measuring stage and sample-loading mechanism: (**a**) overall view and (**b**) partial enlargement. 1—manual fine adjustment of the lift; 2,7—force sensors; 3—sample-loading mechanism; 4—temperature sensors; 5—samples tested; 6—fastening screws.

A set of manual fine-tuning lifts is provided underneath each of the two separate sample-loading mechanisms. This manual fine-adjustment lifting table allows the height of each of the two samples to be adjusted independently and slightly to accommodate the thickness deviation of the two different samples. This ensures that the grinding surfaces of the two samples under test are in the same plane, as shown in Figure 9a. Below these two sets of manual trim lifts there is an automatic lifting mechanism, as shown in Figure 9b.

The automatic lifting mechanism is a modification of an existing height-adjustable optical system platform, whose height adjustment is motor-driven, which can simultaneously control two sets of manually fine-tuned lifting tables in a rapid manner.

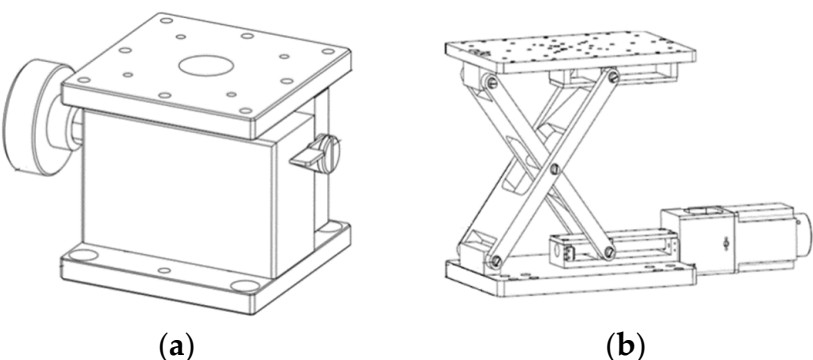

(a)　　　　　　　　　　　　　　(b)

**Figure 9.** Height lift adjustment mechanism: (**a**) manual operation and (**b**) electric controls.

### 4.2. Belt Grinding Mechanism

The belt grinding mechanism can be rotated by 90° under the motor drive, allowing the belt to enter and move away from the surface of the sample. After the test piece has been mounted, the motor-driven belt grinding mechanism reaches the grinding station for the friction and wear test, as shown in Figure 5a. When the grinding work has been completed, the motor drives the belt grinding mechanism to lift up, as shown in Figure 10. This means that the motor actuator support (7), together with the grinding mechanism, moves to a fixed position after rotating 90° along the guide pin (5) around the rotating shaft (6).

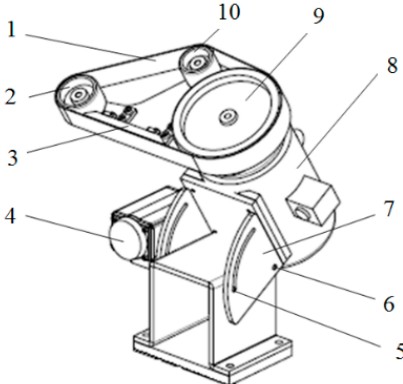

**Figure 10.** Axonometric view of the belt grinding mechanism. 1—sanding belts; 2,9,10—sanding belt support wheels; 3—sanding belt support frame; 4—electric motors; 5—guide pins; 6—spindle; 7—support frame for electric motor actuators; 8—electric motor actuators.

### 4.3. Wear Measurement Mechanism

The device uses in situ length measurement to determine the amount of wear. A schematic diagram of the height measurement of the sample being tested is shown in Figure 11. Before conducting the friction experiment, the grinding tool was first started to make it rotate at low speed, and the surface of the sample under test was levelled according to the pressure sensor below the two samples under test. After the surfaces of the two samples under test were levelled, the abrasive tool was made to leave the surface under test. At this time, the rotating shaft with the high-precision displacement sensor was rotated, so that the sensor probe was in contact with the surface of the two measured samples, and the data were collected at this time. After the friction experiment, the displacement transducer's probe was again made to contact the surface of each of the two measured

samples for measurement. The difference between the measured sample before and after the friction measured by the displacement sensor is the amount of material wear.

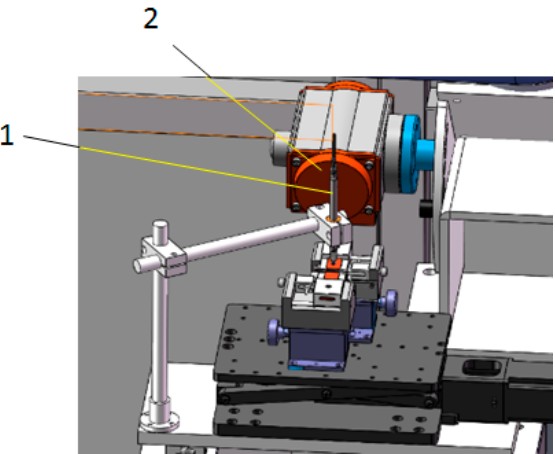

**Figure 11.** Schematic diagram of the height measurement of the sample under test. 1—displacement sensors; 2—data-acquisition equipment.

The wear resistance of the material being tested is reflected by the use of a high-precision displacement sensing probe (1) to monitor the change in height of the test piece before and after the friction test. The probe underneath the displacement sensor (1) is pneumatically controlled to retract so that the probe does not come into direct contact with the sample being tested during positioning of the sensor. Once the displacement sensor (1) has reached the measuring station, the probe on the displacement sensor extends and comes into contact with the surface of the sample being tested. Once the probe underneath the displacement sensor (1) has touched the surface of the sample to be measured, the data-acquisition device starts reading the data. After the data have been processed, the change in thickness of the measured sample is obtained, i.e., the amount of wear.

*4.4. Feasibility Analysis*

The experimental flow diagram of the new friction and wear tester is shown in Figure 12.

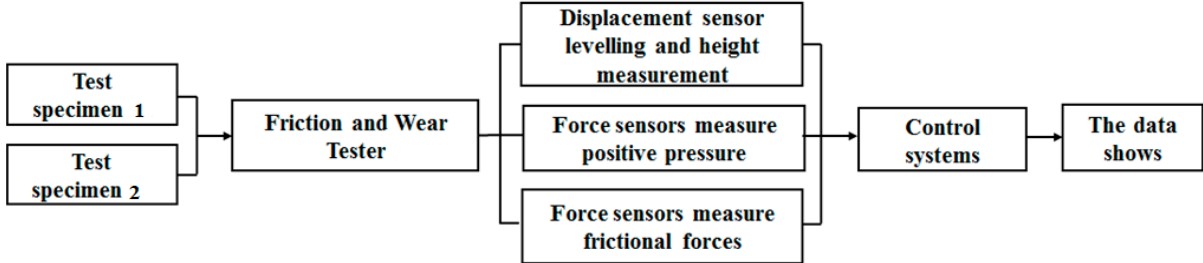

**Figure 12.** Experimental flow diagram of the new friction and wear tester.

Based on the friction testers currently available on the market, our design solution is feasible. There are currently friction testers on the market that determine the wear resistance of the material being tested by measuring the amount of change in length. However, existing friction testers can only test the abrasion resistance of one material at a time. If the wear resistance of two different materials is to be compared, only two tests can be carried out using existing friction testers. This makes it difficult to ensure that the test conditions are identical, and the resulting test errors are relatively large relative to each other, so existing friction testers are not suitable for comparing the friction properties of two different materials with similar wear resistance. The newly designed friction tester enables

the in situ measurement of two samples by measuring the amount of change in height before and after the friction test as an indicator of the wear resistance of the material under test. That is, the amount of change in length is used as the amount of wear of the material.

There are three main components that affect the accuracy of the measurement results when using this equipment for wear resistance measurements. The first aspect is the error in the adjustment of the level of the measured friction surface of the two samples being measured prior to the friction experiment. The second is the error due to the rotation of the spindle on which the displacement transducer is mounted, as shown in Figure 8. There are various methods of levelling horizontal surfaces. Here, the levelling of the measured sample surface is determined by the amount of pressure generated by the abrasive tool against the surface of the measured sample. The main influence on the accuracy of the test is the sensitivity of the pressure transducer. The second is the accuracy of the displacement transducer that measures the change in height of the sample being measured. Grating displacement sensors can be used, which can measure to an accuracy of nanometres. The third is the angular error caused by loading the card and driving the column of the displacement transducer to rotate. However, the angle of rotation that affects the accuracy of the measurement is the angle of rotation between the two samples to be measured, which does not exceed $30°$. Here the option of using high-precision ball bearings is available to reduce angular errors. In addition, the newly designed friction tester with temperature sensors on each side of the mounted sample allows for the temperature of both samples to be observed in real time during the friction test.

### 5. Control Systems and Software Systems

The test system measures wear, friction, load and temperature in accordance with the requirements of the friction test. There are two sets of pressure sensors and one set of thermocouple temperature sensors for each sample being measured. A thermocouple mounted on the table where the test specimen is mounted allows for the temperature change of the specimen due to grinding to be captured at any time. High-precision force sensors are located underneath and to the side of the test piece, allowing for the pressure downwards and the friction to the side of the test piece to be sensed at all times. During the grinding process, temperature and force data are recorded in real time. The collected electrical signals are transmitted via signal transmission lines to the computer control system, then processed and analysed accordingly and displayed on the control system interface in the form of a curve. The control system mainly regulates the starting and stopping status and motor speed of the test machine and data acquisition, as well as monitoring and analysing the status of the test parameters, etc. It mainly consists of a computer, control circuit board and control software. The electrical signal measured by the sensor is converted, amplified and connected to the computer via an A/D converter. Data acquisition and processing is carried out by a dedicated software system developed in-house.

### 6. Conclusions

The amount of wear is an important parameter in evaluating the wear resistance of a material. The accurate and quantitative measurement of wear is an interesting task for the study of the tribological properties of thin-film materials. The new film friction and wear tester developed in this paper can accurately test wear. Compared to various friction testers available today, this device has the following features:

(1) The wear resistance of the material is determined by an in situ comparison of the change in height of the sample under test before and after the friction test, enabling in situ measurement of the amount of wear and tear with high test accuracy. It is particularly suitable for the accurate comparison of small differences in the wear resistance of two different film materials.

(2) The tester can collect the changes in temperature, pressure and friction force of two different film materials during the friction test in real time, providing technical support for the study of the friction and wear mechanism of two different film materials.

**Author Contributions:** Conceptualization, M.L.; Data curation, D.W.; Funding acquisition, M.L.; Investigation, D.W. and M.L.; Methodology, D.W. and M.L.; Writing—original draft, D.W. and M.L. All authors have read and agreed to the published version of the manuscript.

**Funding:** This research was funded by the National Science Foundation of China, Grant No. 11272231, and the Tianjin Natural Fund Committee, Grant No. 20YDTPJC00080.

**Institutional Review Board Statement:** Not applicable.

**Informed Consent Statement:** Not applicable.

**Data Availability Statement:** Not applicable.

**Acknowledgments:** The author would like to thank Industry- University-Research Office of Tianjin University of Commerce, China, for the financial support and assistance and thank Zibo City, Shandong Province, China for the technical guidance given to our work here by Dao Xianwei Intelligent Technology Co.

**Conflicts of Interest:** The authors declare no conflict of interest. The funders had no role in the design of the study; in the collection, analyses, or interpretation of data; in the writing of the manuscript; or in the decision to publish the results.

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
