# Peer review of "Design of a Tester for In Situ Simultaneous Measurement of the Wear of Two Different Film Materials"

_coatings, doi:10.3390/coatings12091359_

Round 1
Reviewer 1 Report (Previous Reviewer 2)
The manuscript was extensively revised by the authors. It might be acceptable as it is now.
Author Response
Please see attachment.

Reviewer 2 Report (New Reviewer)
Dear Authors
In the submitted manuscript the construction of the tester for measuring the wear of thin-film samples were presented. Measurement of two samples simultaneously is possible in described solution. The problem of wear resistance is undoubtedly very important, and any measuring devices for estimating the wear in controlled conditions are a significant support for technological processes. The main advantages of the manuscript is the presentation of the tester, where mounting mechanism, precision measuring stage, belt grinding mechanism, wear measurement mechanism were described in detail. The machine is also equipped with a control system that includes pressure and temperature sensors.
The main weakness of the article is the lack of any results of sample tests.
Section 1 highlights measurable material wear parameters such as change in length, weight, or volume. Unfortunately, the article does not present any sample tests of the device's operation. It is worth to present the operation of the tester in practice, focusing on mentioned nickel-nanodiamond composite layers. In particular, the comparison of testing two different samples should be emphasized. In this case, typical operating parameters should be given: pressure, time, speed, temperature, etc. Additional advantage would be to compare the results with other wear test methods.
In section 2 the problem is describing by presenting the disadvantages of previously used way of conducting wear tests. More description of the experimental-setup used to get the results shown in Figures 2 and 3 is recommended to present in this section.
Furthermore, the manuscript is poor (stylistically and linguistically) and should be somewhat rewritten.
Therefore, in my opinion, the article is incomplete, because there are no research results confirming the effectiveness and advantage of the presented solution over competitive solutions, or at least relating to other methods of wear measuring.
Author Response
Please see attachment.

Reviewer 3 Report (New Reviewer)
This paper reports about the detailed description of one equipment capable to measure two samples simultaneously. Most of the paper is related to the design of the equipment with a detailed description of their parts and operation.
This equipment is interesting because the dual measurement mode of samples and also because the high resolution of 0.1 microns to determine the horizontal leveling of the surfaces.
Authors should address the following points to improve the quality of the paper:
1. The English of the manuscript has some typos which must be corrected.
2. This reference include information relevant for this paper:
J Sol-Gel Sci Technol (2010) 54:312–318 DOI 10.1007/s10971-010-2196-7
3. Authors mention in the manuscript the application of the tester to nickel-nanodiamond composite coatings. However, the results of the wear tests are not included in the paper, authors should include these results.
4. Given the high resolution of the wear tester, how are handled the results for thin film materials with high roughness? Also, although the samples were not rough, the wear on their surface can produce surface roughness leading to the same question?
Author Response
Please see the attachment.

Reviewer 4 Report (New Reviewer)
Review of the manuscript "Design of a tester for in-situ simultaneous measurement of the wear of two different film materials". It is about a novel method to perform friction tests and in-situ measurements of wear on two different materials at the same time.
The manuscript would be original and interesting, however the proposed solution is only from a theoretical point of view. In the whole manuscript only schematic diagrams, renderings and sketches are presented.
Not a prototype has been built!
Not a functional test has been performed!
No experimental results have been presented and discussed!
Furthermore Fig. 11 is not called in the main text.
References (all) do not meet the journal format.
This is not a research article!
The manuscript could be reconsidered for publication only after a deep revision, starting from the contents.
Round 2
Reviewer 2 Report (New Reviewer)
Dear Authors
Thank you for correcting the manuscript and for the provided explanations. Assuming the tester is in the design phase, no comparative results can currently be required. In this context, I believe the article is worth publishing, after the improving the language style.
Reviewer 4 Report (New Reviewer)
The manuscript has been consistently improved and can now be accepted in the present form
This manuscript is a resubmission of an earlier submission. The following is a list of the peer review reports and author responses from that submission.
Round 1
Reviewer 1 Report
The paper is interesting because it proposes a new design of a tester for in-situ simultaneous measurement of the wear of two different film materials. However, the authors have not shown a any experiments and data point collected from this new device and compared with known measuring techniques to justify their claims in conclusion. This paper needs proof of concept and I strongly recommend authors to revisit this paper design and methodologies. Authors must run experiments and prove that this device work and discuss the results in details.
Author Response
Your comments and suggestions have been very helpful to us in our research work here. Work on the design of the friction and wear tester here began in 2015. So far, we have made several iterations of the design of the device, including three major revisions to the mechanical part. Because of the relatively high precision required for the mechanical part of the tester, the cost of developing the device is also relatively high. Due to a lack of research funding, a certain amount of funding is still needed to support subsequent work after the mechanical part of the equipment has been completed. We look forward to research institutes and companies joining us in our research and development work if required.
Our design solution is feasible. This is because the experimental equipment currently in use for determining the wear resistance of materials by measuring the amount of change in length exists. However, all the experimental equipment can only test one sample under the same conditions. To compare the wear resistance of two different materials, two experiments would be required. The measurement error due to different test conditions is relatively large.
To make up for the shortcomings of our work here, in section 2.4 of the revised version of the thesis, we add a feasibility analysis,and is highlighted in yellow.
Reviewer 2 Report
Please check the attached doc.

Reviewer 3 Report
Recommendation: Major revisions are needed.
Comments:
The paper by Wang et al. contributes to the aspect of studying the frictional wear mechanism of materials. The title and abstract are appropriate for the content of the text. Moderate English changes are required. The article gives an interesting historical and scientific perspective on the new design of the friction and wear tester.
Some issues should be addressed before publication.
1. Abstract. Please rephrase the abstract. A lot of sentences here are duplicates with the same meaning. Also, there are two periods at the end of the abstract.
2. Page 1, line 28. “The amount of material lost due to wear is called wear.” Please rephrase. This does not make sense and please add a reference to the definition.
3. Page 1, line 39, 43. Many references are needed here.
4. Page 2, line 50. Do you mean “also in this case”?
5. Page 2, line 59. Reference.
6. Page 2, line 66. “As nano-friction continues to develop, the demands on tribological testing techniques are increasing. It is particularly important to develop and design precision friction test equipment.” These two sentences are duplicated.
7. Page 2, line 70-82. These few lines have a large number of repeat sentences.
8. Page 6, line 195. Please rephrase.
This work is more suitable for a journal that studies machine design or mechanical engineering. Moderate English changes are required for publication.
Author Response
Dear Reviewer,
Your suggestions for our thesis are very good and pertinent.
Your recognition and acknowledgement of our work is a great inspiration to us.
Here are the changes we have made to address the specific issues you have raised.
- Please rephrase the abstract. A lot of sentences here are duplicates with the same meaning. Also, there are two periods at the end of the abstract.
Answer 1: The suggestions you made are very good. We rechecked the abstract section of the paper and have made changes to the abstract. The revised section is highlighted.
- Page 1, line 28.“The amount of material lost due to wear is called wear.” Please rephrase. This does not make sense and please add a reference to the definition.
Answer 2: In the revised version, we have replaced the original statement with "The wear resistance of a material is measured by the amount of wear".
- Page 1, line 39, 43. Many references are needed here.
Answer 3: We take your advice. We are reworking this section in the revised manuscript and adding references that we have previously checked.
- Page 2, line 50.Do you mean “also in this case”?
Answer 4: You are correct in your understanding. It has been amended in the revised draft.
- Page 2, line 59.
Answer 5: In the revised version, the references have been added.
- Page 2, line 66.“As nano-friction continues to develop, the demands on tribological testing techniques are increasing. It is particularly important to develop and design precision friction test equipment.” These two sentences are duplicated.
Answer 6: We have made changes here in the revised version.
- Page 2, line 70-82.These few lines have a large number of repeat sentences.
Answer 7: In the revised version of the thesis we have reworked this part of the expression.
- Page 6, line 195. Please rephrase.
Answer 8: The expression in this section has been revised in the revised draft.
We submitted this paper to the journal “Materials” for two reasons. The journal's processing of manuscripts is fast.Second, the new testers we have developed are used to test the abrasion resistance of materials and have a strong relationship with the material.
Round 2
Reviewer 2 Report
The authors' have amended the manuscript adequately. It could be accepted.
Reviewer 3 Report
Accept in present form